}RESEARCH ARTICLE

# Does perioperative respiratory event increase length of hospital stay and hospital cost in pediatric ambulatory surgery?

**Maliwan Oofuvong**[1]*, **Alan Frederick Geater**[2©], **Virasakdi Chongsuvivatwong**[2©], **Thavat Chanchayanon**[1], **Bussarin Sriyanaluk**[1], **Boonthida Suwanrat**[1], **Kanjana Nuanjun**[1]

**1** Department of Anesthesiology, Faculty of Medicine, Prince of Songkla University, Hat Yai, Songkhla, Thailand, **2** Epidemiology Unit, Faculty of Medicine, Prince of Songkla University, Hat Yai, Songkhla, Thailand

© These authors contributed equally to this work.

* oomaliwa@gmail.com

## Abstract

### Objective

We examined the consequences of perioperative respiratory event (PRE) in terms of hospitalization and hospital cost in children who underwent ambulatory surgery.

### Methods

This subgroup analysis of a prospective cohort study (ClinicalTrials.gov: NCT02036021) was conducted in children aged between 1 month and 14 years who underwent ambulatory surgery between November 2012 and December 2013. Exposure was the presence of PRE either intraoperatively or in the postanesthetic care unit or both. The primary outcome was length of stay after surgery. The secondary outcome was excess hospital cost excluding surgical cost. Financial information was also compared between PRE and non-PRE. Directed acyclic graphs were used to select the covariates to be included in the multivariate regression models. The predictors of length of stay and excess hospital cost between PRE and non-PRE children are presented as adjusted odds ratio (OR) and cost ratio (CR), respectively with 95% confidence interval (CI).

### Results

Sixty-three PRE and 249 non-PRE patients were recruited. In the univariate analysis, PRE was associated with length of stay (p = 0.004), postoperative oxygen requirement (p <0.001), and increased hospital charge (p = 0.006). After adjustments for age, history of snoring, American Society of Anesthesiologists physical status, type of surgery and type of payment, preoperative planned admission had an effect modification with PRE (p <0.001). The occurrence of PRE in the preoperative unplanned admission was associated with 24-fold increased odds of prolonged hospital stay (p <0.001). PRE was associated with higher excess hospital cost (CR = 1.35, p = 0.001). The mean differences in contribution margin for total procedure (per patient) (PRE vs non-PRE) differed significantly (mean = 1,523; 95% CI: 387, 2,658 baht).

**Data Availability Statement:** All relevant data are within the manuscript and its Supporting Information files.

**Funding:** The authors received no specific funding for this work.

**Competing interests:** The authors have declared that no competing interests exist.

## Conclusion

PRE with unplanned admission was significantly associated with prolonged length of stay whereas PRE regardless of unplanned admission increased hospital cost by 35% in pediatric ambulatory surgery.

## Trial registration

ClinicalTrials.gov registration number NCT02036021.

## Introduction

Ambulatory pediatric surgery can shorten hospital stay, reduce risk of nosocomial infections, and reduce hospitalization costs [1]. A perioperative respiratory event (PRE) such as laryngospasm, bronchospasm, and desaturation in pediatric anesthesia is not uncommon, especially in high-risk children (age < 3 years, recent upper respiratory tract infection, history of rhinitis, habitual snoring, obesity) [1–4] or those who have certain types of surgery such as airway surgery and adenotonsillectomy [5, 6]. Edler et al. [6] reported prolonged stay in the post-anesthetic care unit (PACU) by comparing patients who had PRE with patients without PRE in pediatric ambulatory tonsillectomy. In our previous study [7] we reported that the occurrence of PRE prolonged the length of stay and increased both direct hospital cost and indirect cost such as transportation and parental loss of income. However, the majority of subjects in our previous study were inpatients. In the current study, we performed a secondary analysis confined to only ambulatory surgery patients using the data from our previous study regarding the effects of PRE on excess hospital cost. The study was registered at ClinicalTrials.gov: NCT02036021.

## Materials and methods

This subgroup analysis of a prospective cohort study was approved by the Human Research Ethics Committee, Faculty of Medicine, Prince of Songkla University (Chairperson Assoc. Prof. Boonsin Tangtrakulwanich) on 3 March 2021 (REC 64-086-8-1). This study was part of a larger research project (ClinicalTrials.gov: NCT02036021). Children aged between 1 month (term infants) and 14 years who underwent general anesthesia (GA) for ambulatory surgery between November 2012 and December 2013 were included. Written informed consent was obtained from all parents. The patients who developed PRE (PRE group) were compared with a control group who did not have any PRE (non-PRE group) in terms of length of hospital stay postoperatively and excess hospital cost. Excess hospital cost was defined as hospital direct cost that did not include surgical costs. The DOI link by Protocols.io is dx.doi.org/10.17504/protocols.io.bt6vnre6.

### Hospital cost system and length of stay

Costs and length of hospitalization were retrieved from the hospital information system. Since there is no system of cost unit analysis in our hospital, the hospital charge was used to represent direct hospital cost (the hospital charge multiplied by cost-to-charge ratio) [8, 9]. Subsequently, a fixed cost-to-charge ratio eventually cancelled out when the hospital cost in PRE was divided by the hospital cost in non-PRE. Since we focused on direct hospital cost in

pediatric ambulatory surgery, the indirect costs (the combination of transportation cost and parental loss of income) were omitted in the analysis.

## Participants

Children were included if they fit the criteria for ambulatory surgery. They were excluded if a written informed consent could not be obtained from the parents.

## Main exposure (PRE and non-PRE)

In our hospital, all patients are monitored under anesthesia surveillance using continuous pulse oximetry, capnography and electrocardiography incorporating the vital signs every 5 minutes. The PRE group was defined as children who had perioperative respiratory events such as laryngospasm, bronchospasm, upper airway obstruction [10], or reintubation either intraoperatively or in the PACU period with or without having desaturation. Desaturation was defined as oxygen saturation ($SpO_2$) by pulse oxymetry that was <95% for more than 10 seconds [11]. The occurrence of PRE, causes of PRE, and the lowest $SpO_2$ intraoperatively or at PACU were recorded immediately in the vital signs table and in the data record form by the anesthetist nurse in charge of each operating theater. The patients with PRE and the lowest recorded $SpO_2$ were placed into 3 categories based on the occurrence/severity of perioperative desaturation (PD): no PD ($SpO_2$ >94%), mild to moderate PD ($SpO_2$ 86–94%), and severe PD ($SpO_2$ <86%). Patients in the non-PRE group were defined as children who did not develop any PRE intraoperatively or in the PACU period based on the recorded data form. Occurrence/severity of PRE was divided into 3 categories: non-PRE, mild to moderate PRE ($SpO_2$ 86–99%) and severe PRE ($SpO_2$ <86%). Time to first PRE event and duration of PRE were also recorded to increase the accuracy of the main exposure.

## Outcomes of interest

**Prolonged hospitalization post-surgery.** The primary outcome was length of stay post-surgery. Any hospital stay recorded by the PACU nurses followed approval by both the surgeon and anesthesiologist in charge. The number of days of hospitalization post-surgery as well as occurrence of postoperative complications were obtained from the hospital information system by the principal investigator (MO). In our hospital, 25% of ambulatory surgery cases are planned admissions, usually occurring from surgical concerns or parent's preference (insurance/distance to hospital). An unplanned admission could arise from an anesthetic adverse event or surgical complication. Based on past data, the median length of stay for a planned admission was 1.0 day. Therefore, prolonged length of stay post-surgery was defined as the number of hospitalization days more than 1 day for a planned admission and at least 1 day for an unplanned admission post-surgery. The duration of PACU stay and postoperative oxygenation were recorded by the PACU nurses.

**Excess hospital cost.** The secondary outcome was excess hospital cost. Hospital charge was used instead of direct hospital cost for a comparison between PRE and non-PRE. Thus, excess hospital charge was defined as all hospital charges excluding the surgical charge in the PRE group minus those in the non-PRE groups. Hospital charges included the use of resources within the health sector, e.g. home medication, anesthesia charge, and hospitalization [12]. After the patient was discharged, total hospital charges were obtained from the hospital information system and recorded by the principal investigator (MO).

## Financial information

In the area of hospital planning, financial information including net revenue, direct hospital costs, fixed costs, and variable costs need to be addressed. Gross revenue arises from the hospital charges of each outpatient, or inpatient if admitted. We used a cost-to-charge ratio of 0.4 based on our previous estimate of direct hospital costs [7], therefore, direct hospital costs were calculated from hospital charges multiplied by 0.4. Since fixed expenditures associated with buildings, salaries, equipment and other overhead were not obtained from our previous study [7], fixed cost was omitted in the present study. Thus, the variable "costs" included medication and supplies, which change based on the number of patients treated, which were obtained from the hospital information system [13]. Therefore, direct hospital cost, e.g. accommodation, meals, medication, laboratory, and nursing care service, would represent the variable cost in our setting. Since contribution margin represents actual net cash flows for individual patients in terms of delivery of care [14], the contribution margin instead of total margin was calculated in this study. The contribution margin was obtained from gross revenue (hospital charges) minus variable costs (direct hospital cost). Therefore, the contribution margin in our setting was calculated from hospital charges multiplied by 0.6.

## Potential confounding variables

Patient-related characteristics and type of payment system were obtained at the preoperative period by the investigative team (BS, BS, KN) while surgical and anesthesia-related variables were obtained at the intraoperative period by the anesthetist nurse in charge of each operating theater. Patient-related characteristics included age, sex, body mass index (kg/m$^2$), history of upper respiratory tract infection [7], obesity (>95 percentile weight for age), and history of snoring. Surgical and anesthesia-related data included type of surgery, American Society of Anesthesiologist (ASA) classification, choice of anesthesia, technique of anesthesia, induction agent, intubating agent, inhalation agent, gas mixed with oxygen, and narcotic use. Type of preoperative admission (planned vs unplanned), which was decided by the surgeon, was also included as a potential confounding variable.

## Statistical analysis

Data record forms were created and information was abstracted from the electronic medical records and double-entered into a database using EpiData version 3.1. R software was used to analyze the data (R version 4.0.2, R Core Team, Vienna). Descriptive statistics including frequency with percentage and mean with standard deviation (SD) or median with interquartile range as appropriate in the PRE and non-PRE groups were computed. Continuous variables for normally or non-normally distributed variables were compared using the unpaired Student's t-test or Wilcoxon's rank sum test, respectively. The chi-square test or Fisher's exact test was used to compare categorical variables. To compare the main outcomes between two groups, comparisons on the outcomes of interest were adjusted for potential confounders using logistic regression models.

**Model for prolonged hospitalized post-surgery.** The association between the main exposure (PRE vs non-PRE) and prolonged length of hospital stay post-surgery was determined by cross-tabulation. A directed acyclic graph (DAG) was used to represent the potential causal relationships among the covariates (including PRE) and the outcomes using DAGitty software version 3.0. Potential confounding variables including hospital payment suggested by the DAG were then selected for a multivariate logistic regression model and were retained in the model irrespective of their statistical significance [15, 16]. The association between prolonged

length of hospital stay post-surgery and PRE is presented as an adjusted odds ratio (OR) with 95% confidence interval (CI).

**Model for adjusted excess hospital cost.** A DAG was also used to represent the potential causal relationships among covariates (including PRE) and excess hospital charge. To model the relationship between PRE and excess hospital cost, potential covariates indicated by the DAG, including preoperative planned admission and the type of hospital payment system, were included in a multiple linear regression model. To fit the residual of linear distribution assumption, the so-called adjusted excess charge obtained by the log of excess charge more than 2,000 baht was used for the final excess hospital cost parameter. The exponentials of their coefficients (cost ratio [CR] and 95% CI) were displayed and considered significant if the F test p values were <0.05.

To further determine the impact of the severity of PRE on hospital stay and hospital cost, PRE was replaced with a variable indicating severity of PRE after obtaining the final model. The effect modification between the potential predictors and PRE/severity of PRE on the outcomes were evaluated for each final model.

## Sample size calculation

For the primary outcome, the proportion of prolonged hospital stay was estimated from our previous study which reported that 39% of the PRE group and 18% of the non-PRE group were post-surgery admissions in outpatient surgery [7]. Since the incidence of PRE was estimated as 17–20%, at least 40 PRE children and 200 non-PRE children were required to detect a difference in these proportions under a power of 80% and type I error of 5%. For the secondary outcome, the means and standard deviations of the log excess hospital costs (in baht) between PRE (9.94 ± 0.90) and non-PRE (8.62 ± 0.85) was estimated from our previous study [7]. At least 43 PRE children and 215 non-PRE children were required to detect a difference in these magnitudes under a power of 80% and type I error of 5%. Therefore, at least 48 children in the PRE group and 240 children in the non-PRE group were required with compensation for a 10% drop-out rate. Fourteen months of data collection from outpatient surgery patients contained 63 children in the PRE group and 249 children in the non-PRE group, which adequately met the required sample size.

## Results

Informed consent was obtained from a total of 312 out of 428 eligible children from November 2012 to December 2013 at Songklanagarind Hospital (Fig 1). Table 1 shows the characteristics by severity of PRE and indicates that the two most common types of PRE were desaturation (40 events, 63.5%) and upper airway obstruction (8 events, 12.7%). Forty events (63.5%) occurred in the intraoperative period. Fourteen (22%) and 44 children (70%) had severe PRE (SpO$_2$ < 86%) and mild to moderate PRE (SpO$_2$ 86–99%), respectively. Table 2 compares baseline demographic data and respiratory- and anesthesia-related variables in children with and without PRE. Baseline characteristics were well balanced in their baseline characteristics between the groups except for ASA classification (p = 0.028). The proportion of patients who had ASA classification 2 was higher in the PRE group (76%) than those in the non-PRE group (58%). Considering ASA 1 and 2 were healthy and mild systemic disease patients, ASA classification 3 was quite balanced between the PRE (4.8%) and the non-PRE groups (5.6%). The major hospital payment system was universal coverage, which accounted for 67% in the PRE group and 69% in the non-PRE group. Since the majority of PRE occurred during the intraoperative period, anesthetic duration was considered as the consequence of PRE and was categorized as an outcome of the study. Fig 2 shows the distribution of number of days of

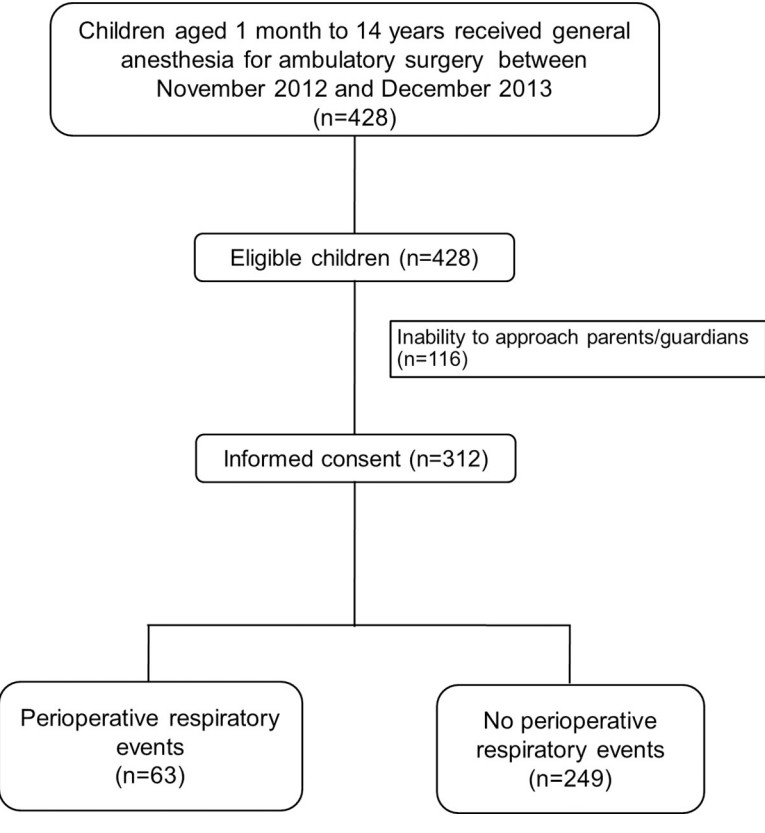

**Fig 1. Flow diagram of the study.**

hospitalization post-surgery between non-PRE and PRE groups. Approximately 40% of children in the PRE group were admitted for at least 1 day.

Table 3 shows the outcomes of interest in children with and without PRE. Compared to non-PRE children, PRE children had a higher proportion who required a postoperative

Table 1. Characteristics and severity of perioperative respiratory events.

| Characteristic | Severity | | | p-value |
|---|---|---|---|---|
| | Mild (SpO$_2$ > 94%) (*n* = 5) | Moderate (SpO$_2$ 86–94%) (*n* = 44) | Severe (SpO$_2$ < 86%) (*n* = 14) | |
| **Event** | | | | <0.001 |
| Desaturation | 0 (0) | 34 (77.3) | 6 (42.9) | |
| Stridor | 4 (80) | 2 (4.5) | 2 (14.3) | |
| Bronchospasm | 1 (20) | 3 (6.8) | 1 (17.1) | |
| Laryngospasm | 0 (0) | 3 (6.8) | 2 (14.3) | |
| Endotracheal tube displacement | 0 (0) | 2 (4.5) | 3 (21.4) | |
| **Period of event** | | | | 0.004 |
| At intraoperative only | 2 (40) | 26 (59.1) | 8 (57.1) | |
| At PACU only | 3 (60) | 18 (40.9) | 2 (14.3) | |
| Both | 0 (0) | 0 (0) | 4 (28.6) | |
| Time to event (min) | 80 [70, 90] | 53 [10, 125] | 55 [19, 74] | 0.598 |
| Duration of event (min) | 5.0 [1.0, 5.0] | 1.0 [1.0, 2.0] | 1.0 [1.0, 4.2] | 0.599 |
| Range (min) | 1.0–70.0 | 1.0–50.0 | 1.0–35.0 | |

**Note**: The values are presented as the number of patients (%) per group or the median [interquartile range]. SpO$_2$, Oxygen saturation; PACU, postanesthesia care unit.

**Table 2. Comparison of characteristics of children having general anesthesia with and without PRE.**

| Variables | PRE (n = 63) | Non-PRE (n = 249) | p-value |
|---|---|---|---|
| **Demographic data** | | | |
| Age (months) | 44 [29.5, 92.0] | 60.5 [37.0, 92.5] | 0.112 |
| Age (months) | | | 0.965 |
| < 12 | 3 (4.8) | 10 (4.0) | |
| 12–72 | 38 (60.3) | 152 (61.0) | |
| > 72 | 22 (34.9) | 87 (34.9) | |
| Male | 42 (66.7) | 180 (72.3) | 0.469 |
| Anthropometry | | | |
| Weight (kg) | 14.0 [11.8, 22.7] | 16.8 [13.0, 22.6] | 0.137 |
| Height (cm), mean (SD) | 104.2 (23.7) | 107.8 (21.4) | 0.238 |
| Body mass index (kg/m$^2$) | 15.7 [14.4, 18.6] | 15.1 [13.7, 17.4] | 0.089 |
| Obesity | 10 (15.9) | 26 (10.4) | 0.325 |
| Respiratory-related variables | | | |
| Upper respiratory tract infection | 11 (17.5) | 24 (9.6) | 0.125 |
| Smoking | 32 (50.8) | 125 (50.2) | 1.00 |
| Snoring | 18 (28.6) | 75 (30.1) | 0.931 |
| **Surgical and anesthesia-related variables** | | | |
| Type of surgery/procedure | | | 0.723 |
| Urology | 25 (39.7) | 108 (43.4) | |
| Ear-nose-throat | 17 (27.0) | 51 (20.5) | |
| Eye | 10 (15.9) | 46 (18.5) | |
| Others | 11 (17.5) | 44 (17.7) | |
| Preoperative planned admission | 19 (30.2) | 53 (21.3) | 0.185 |
| ASA classification | | | 0.028 |
| 1 | 12 (19.0) | 90 (36.1) | |
| 2 | 48 (76.2) | 145 (58.2) | |
| 3 | 3 (4.8) | 14 (5.6) | |
| Type of anesthesia | | | 0.932 |
| GA alone | 40 (63.5) | 161 (64.7) | |
| GA with caudal block | 16 (25.4) | 58 (23.3) | |
| GA with peripheral nerve block | 7 (11.1) | 30 (12.0) | |
| Airway management | | | 0.427 |
| Mask | 27 (42.9) | 87 (34.9) | |
| Laryngeal mask airway | 8 (12.7) | 44 (17.7) | |
| Endotracheal tube | 28 (44.4) | 118 (47.4) | |
| Induction agent | | | 0.754 |
| Sevoflurane | 55 (87.3) | 211 (84.7) | |
| Propofol | 8 (12.7) | 38 (15.3) | |
| Intubating agent | | | 0.403 |
| Succinylcholine | 3 (4.8) | 7 (2.8) | |
| NDMR | 17 (2.7) | 46 (18.5) | |
| Sevoflurane | 5 (7.9) | 16 (6.4) | |
| Propofol | 2 (3.2) | 10 (4.0) | |
| No endotracheal tube | 36 (57.1) | 170 (68.3) | |

(*Continued*)

**Table 2.** (Continued)

| Variables | PRE (n = 63) | Non-PRE (n = 249) | p-value |
|---|---|---|---|
| Inhalation agent | | | 0.262 |
| Sevoflurane | 58 (92.1) | 237 (95.2) | |
| Isoflurane/Desflurane | 2 (3.2) | 2 (0.8) | |
| None | 3 (4.8) | 10 (4.0) | |
| Gas mixed with oxygen | | | 0.315 |
| Air | 53 (84.1) | 216 (86.7) | |
| $N_2O$ | 10 (15.9) | 27 (10.8) | |
| 100% $O_2$ | 0 (0.0) | 6 (2.4) | |
| Narcotic | | | 0.764 |
| Intravenous fentanyl | 61 (96.8) | 236 (94.8) | |
| Caudal narcotic | 2 (3.2) | 8 (3.2) | |
| None | 0 (0.0) | 5 (2.0) | |
| Payment system variable | | | |
| Type of payment | | | 0.284 |
| Universal Coverage | 42 (66.7) | 172 (69.1) | |
| CGD | 13 (20.6) | 41 (16.5) | |
| Self-pay | 5 (7.9) | 32 (12.9) | |
| Government corporation | 2 (3.2) | 3 (1.2) | |
| Private insurance | 1 (1.6) | 1 (0.4) | |

**Note**: The values are presented as the number of patients (%) per group and the median [interquartile range] if not stated otherwise, Others = mass excision/ gastroscopy. ASA, American Society of Anesthesiologist; CGD, The Comptroller General's Department; ETT, Endotracheal tube; NDMR, Non-depolarizing muscle relaxant; PNB. Peripheral nerve block; RA, Regional anesthesia; PRE, Perioperative respiratory event.

oxygen device (p <0.001), had a longer anesthetic time (p <0.001), were more likely to require hospitalization post-surgery in both planned and unplanned admissions (p <0.001), and had a longer hospital stay (p = 0.004). Thirty percent of PRE patients and 21% of non-PRE patients were preoperative planned admission but admission among preoperative unplanned admission was found only in the PRE group (13%). Causes of unplanned admission (8 cases) were from upper airway obstruction (3 cases at PACU), hypoventilation (3 cases at PACU), and wheezing (1 case intraoperative and 1 case at PACU). Of the 8 cases of unplanned admission, 3 cases developed severe desaturation which arose from upper airway obstruction (lowest $SpO_2$ = 26%), hypoventilation (lowest $SpO_2$ = 81%), and wheezing (lowest $SpO_2$ = 68%). The cost parameter, i.e. hospital charge and nursing care service, in the PRE group was significantly higher than that in the non-PRE group (p = 0.006 and p = 0.002, respectively) (Table 3).

Table 4 shows net revenue and contribution margin among the types of surgery between the PRE and non-PRE groups. The mean differences in contribution margin (per patient) differed significantly in total procedure (p = 0.009) and in direct laryngoscopy and bronchoscopy surgery (p = 0.04).

Table 5 shows the predictors of excess hospital cost, length of stay and prolonged length of stay. Geometric mean (SD) refers to the exponential of the mean (SD) log of the excess cost, which is compatible with the mean (SD) of the excess cost.

## Effect modification between potential confounding variables and PRE

Twelve variables (age, obesity, history of upper respiratory tract infection, history of snoring, type of surgery, ASA classification, preoperative planned admission, type of anesthesia,

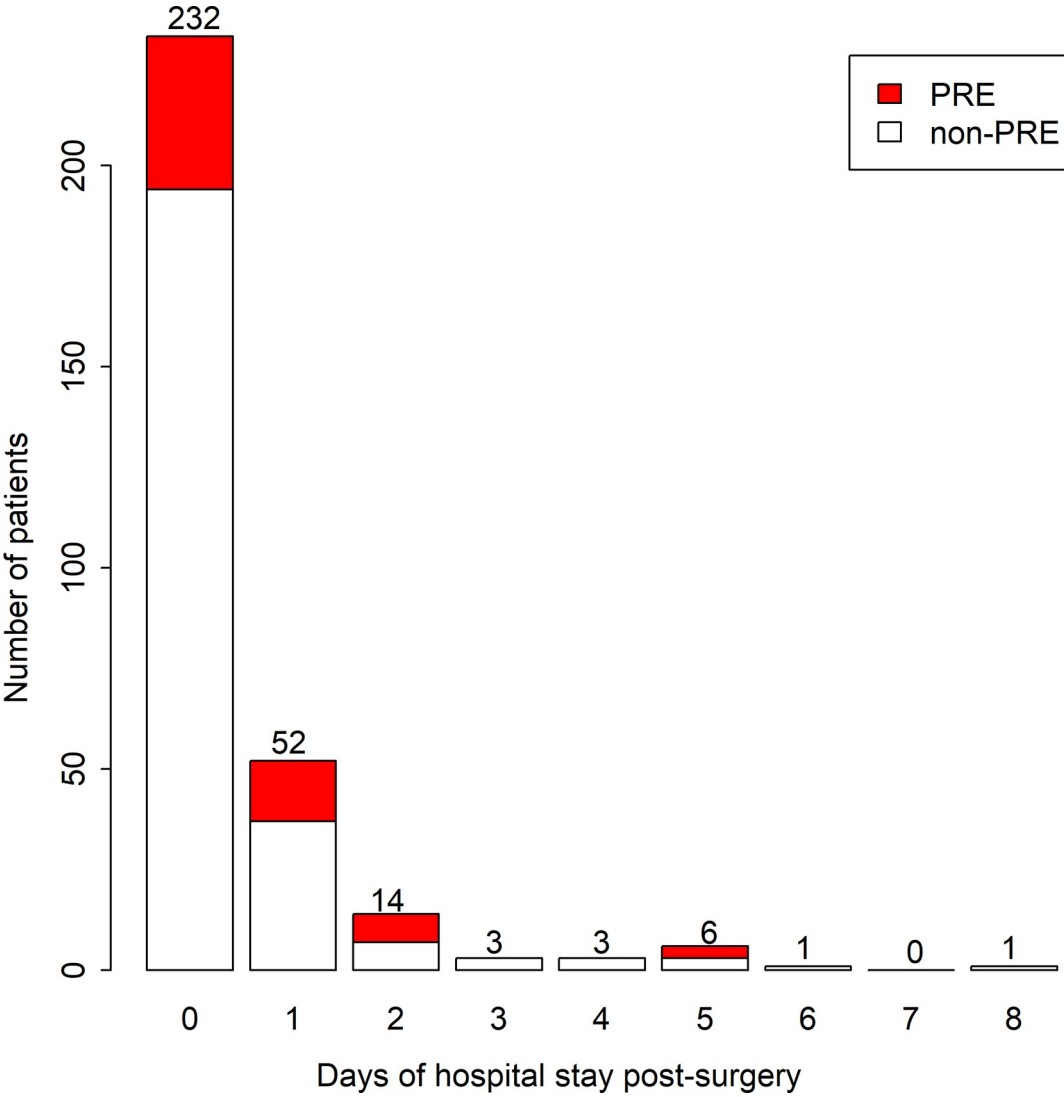

**Fig 2. The number of days of hospitalization post-surgery among non-PRE ($N$ = 249) and PRE group ($N$ = 63). PRE, perioperative respiratory events.** For PRE group, n = 38 in day 0, n = 15 in day 1, n = 7 in day 2, n = 3 in day 5.

induction agent, airway management, type of payment, and PRE) suggested by the previous literature review were related to prolonged hospital stay post-surgery. Exploration of the effect modification between those variables and PRE for prolonged hospital stay revealed only type of preoperative admission (planned vs unplanned) modified the effect of PRE (p <0.001). A cross classification variable between type of admission and PRE/non-PRE was therefore used in the model to estimate the effect on prolonged hospital stay of each combination.

## Analysis of prolonged hospital stay post-surgery

Five potential biasing variables (age, ASA classification, history of snoring, type of surgery, and type of payment) of the total effect of PRE indicated by the DAG (S1 Fig) were included as the minimally sufficient adjustment set with a cross classification variable (Table 6). PRE (vs non-PRE) had no significant effect on prolonged hospital stay among preoperative planned admission (OR = 1.7, 95% CI: 0.5, 5.8). Regardless of having PRE, preoperative unplanned admission

**Table 3. Outcome of interest of children having general anesthesia with and without PRE.**

| Variables | PRE (n = 63) | Non-PRE (n = 249) | Mean differences (95% CI) | p-value |
|---|---|---|---|---|
| Duration of anesthesia (min) | 80 [60, 113] | 65 [40, 85] | 24.2 (8.5 to 39.8) [a] | < 0.001* |
| PACU stay (min) | 70 [50, 100] | 65 [45, 100] | 0.45 (−11.4 to 12.3) | 0.615 |
| Admission outcome | | | | < 0.001** |
| Discharge same day | 36 (57.1) | 196 (78.7) | | |
| Planned admission | 19 (30.2) | 53 (21.3) | – | |
| Unplanned admission | 8 (12.7) | 0 (0) | | |
| Postoperative oxygen device | 5 (7.9) | 0 (0.0) | – | < 0.001** |
| Postoperative fever | 1 (1.6) | 0 (0.0) | – | 0.202 |
| Length of stay (days) | 0 [0, 1.0] | 0 [0, 0] | 0.29 (−0.03 to 0.62) | 0.004* |
| Prolonged length of stay | 15 (23.8) | 20 (8.0) | – | < 0.001** |
| Hospital cost[b]$ | 4,096 [3,320, 6,046] | 3,632 [2,272, 4,795] | 1,015.0 (257.9 to 1772.1)[a] | 0.006* |
| Hospital charge$ (gross revenue) | 10,240 [8,301, 15,115] | 9,080 [5,680, 11,988] | 2,537.5 (644.7 to 4,430.3)[a] | 0.006* |
| Accommodation$ | 0 [0, 300] | 0 [0, 0] | 84.3 (−18.1 to 186.6) | 0.008* |
| Meal$ | 0 [0, 140] | 0 [0, 0] | 52.5 (−13.9 to 118.9) | 0.018* |
| Medication$ | 64 [21, 491] | 49 [17, 302] | 151.3 (−52.0 to 354.6) | 0.095 |
| X-ray and laboratory$ | 0 [0, 220] | 0 [0, 0] | 42.6 (−120.0 to 205.3) | 0.045* |
| Oxygen therapy (includes suction)[§§] | 133.2 (296.6) | 88.3 (208.2) | 44.9 (−34.0 to 123.8) | 0.261 |
| Surgical charge$ | 3,900 [2,700, 4,825] | 3,900 [1,900, 4,500] | 584 (−49.5 to 1218.3) | 0.108 |
| Procedure related to surgery[§§] | 57.3 (290.4) | 38.6 (144.0) | 18.7 (−56.6 to 93.9) | 0.622 |
| Anesthesia charge$ | 1,900 [1,400, 3,075] | 1,400 [1,400, 2,350] | 525.6 (104.9 to 946.3) [a] | 0.006* |
| Procedure related to anesthesia$ | 950 [360, 1,150] | 950 [460, 1,175] | -57.7 (−205.8 to 90.4) | 0.248 |
| Material charge$ | 2,092 [1,115, 3,754] | 1,902 [602, 2,527] | 945.2 (−21.7 to 1,912.1) | 0.089 |
| Nursing care service$ | 50 [50, 500] | 50 [50, 50] | 130.7 (−34.9 to 296.21) | 0.002* |
| Excess hospital charge[c]$ | 6,437 [4,818, 10,157] | 5,279 [3,746, 6,562] | 1953.1 (511.2 to 3395.0) | 0.002* |

**Note**: The values are presented as the median [interquartile range] and the number of patients (%) per group

$ displayed with mean (SD) instead of median [interquartile range] because these variables had $P_{25}$, $P_{50}$, $P_{75}$ being zero

*Wilcoxon rank sum test

**Chi-square test. PRE, perioperative respiratory event; PACU, Postanesthesia care unit.

[a] $p < 0.05$ by unpaired t-test

$31.25 baht per U.S. dollar

[b] calculated from hospital charge $\times$ 0.4

[c] calculated from hospital charge–surgical charge. All costs are in Thai baht.

(vs planned admission) decreased the odds of prolonged hospital stay (p <0.001). However, the occurrence of PRE in a preoperative unplanned admission was associated with a 24-fold (0.26/0.011) increased odds of prolonged hospital stay (p <0.001). After replacing PRE by the severity of PRE, mild to moderate and severe PRE in a preoperative unplanned admission was associated with a 17-fold (0.19/0.011) and 46-fold (0.51/0.011) increased odds of prolonged hospital stay, respectively (p <0.001 and p <0.001, respectively) (Table 6).

## Analysis of adjusted excess hospital cost

Thirteen variables (age, obesity, history of upper respiratory tract infection, history of snoring, type of surgery, ASA classification, type of anesthesia, induction agent, airway management, narcotics use, type of payment, preoperative planned admission and PRE) suggested by the previous literature review were related to excess hospital cost. There was no evidence of effect modification between preoperative planned admission or any other variables with PRE in the

**Table 4. Net revenue and contribution margin among types of surgery between PRE and non-PRE (N = 312).**

| Type of surgery | Without PRE (baht)$ | | | | With ≥ 1 PRE (baht)$ | | | | Difference (PRE vs non-PRE) in contribution margin per patient, baht$ (95% CI) |
|---|---|---|---|---|---|---|---|---|---|
| | No. of patients | Net revenue | Net revenue per patient (95% CI) | Contribution margin[c] per patient (95% CI) | No. of patients | Net revenue | Net revenue per patient (95% CI) | Contribution margin[c] per patient (95% CI) | |
| Minor eye surgery[a] | 28 | 126,093 | 4,503 (3,620 to 5,380) | 2,702 (2,170 to 3,230) | 5 | 23,491 | 4,698 (3,780 to 5,610) | 2,819 (2,270 to 3,370) | 117 (−721 to 955) |
| Eye muscle resection | 13 | 160,663 | 12,359 (11,600 to 13,100) | 7,415 (6,940 to 7,890) | 2 | 26,567 | 13,285 (12,400 to 14,100) | 7,971 (7,470 to 8,470) | 556 (−500 to 1,611) |
| Vitreoretinal surgery | 5 | 121,606 | 24,321 (9,720 to 38,900) | 14,593 (5,830 to 23,400) | 3 | 93,068 | 31,023 (26,100 to 35,900) | 18,614 (15,700 to 21,500) | 4,021 (−8,230 to 16,272) |
| Tongue tie operation | 17 | 80,809 | 4,754 (4,550 to 4,960) | 2,852 (2,730 to 2,970) | 2 | 8,794 | 4,397 (2,310 to 6,490) | 2,638 (2,210 to 3,060) | −214 (−2,263 to 1,835) |
| Tonsillectomy | 4 | 43,639 | 10,910 (9,810 to 12,000) | 6,546 (5,880 to 7,210) | 4 | 42,297 | 10,574 (8,160 to 13,000) | 6,345 (4,900 to 7,780) | −201 (−2,409 to 2,006) |
| Direct laryngoscopy and bronchoscopy | 22 | 280,649 | 12,757 (10,500 to 15,000) | 7,654 (6,310 to 8,990) | 7 | 115,896 | 16,557 (14,100 to 19,100) | 9,934 (8,430 to 11,400) | 2,279 (106 to 4,454)* |
| Ear surgery | 5 | 70,192 | 14,038 (7,350 to 20,700) | 8,423 (4,400 to 12,400) | 4 | 54,975 | 13,744 (2,940 to 24,500) | 8,246 (1,780 to 14,700) | -177 (-10,054 to 9,700) |
| Urethral dilatation | 13 | 57,485 | 4,422 (3,640 to 5,200) | 2,653 (2,190 to 3,120) | 2 | 8,420 | 4,210 (3,090 to 5,330) | 2,526 (1,860 to 3,200) | −127 (−1,799 to 1,545) |
| Minor urologic surgery [b] | 69 | 653,450 | 9,470 (9,070 to 9,870) | 5,682 (5,440 to 5,920) | 16 | 158,622 | 9,914 (8,880 to 10,900) | 5,948 (5,330 to 6,560) | 266 (−438 to 971) |
| Urethroplasty | 27 | 406,301 | 15,048 (13,700 to 16,300) | 9,029 (8,250 to 9,810) | 7 | 120,950 | 17,279 (13,900 to 20,700) | 10,367 (8,330 to 12,400) | 1,338 (−1,245 to 3,921) |
| Excision superficial mass | 30 | 286,588 | 9,553 (7,270 to 11,800) | 5,732 (4,360 to 7,100) | 9 | 102,791 | 11,421 (7,300–15,500) | 6,853 (4,380 to 9,320) | 1,121 (−1,987 to 4,230) |
| Gastroscopy | 16 | 140,082 | 8,755 (7,040 to 10,500) | 5,253 (4,220 to 6,280) | 2 | 18,191 | 9,096 (6,860 to 11,300) | 5,457 (4,110 to 6,810) | 204 (−2,924 to 3,332)) |
| Total | 249 | 2,427,557 | 9,749 (9,050 to 10,400) | 5,850 (5,430 to 6,270) | 63 | 774,064 | 12,287 (10,600 to 14,000) | 7,372 (6,330 to 8,410) | 1,523 (387 to 2,658)** |

$31.25 baht per U.S. dollar

[a]eye examination/change eye prothesis

[b]circumcision/hydrocelectomy/herniotomy

[c]calculated from hospital charge × 0.6

*p <0.05 by unpaired t-test

**p <0.01 by unpaired t-test.

excess hospital cost model. Seven potential biasing variables (age, ASA classification, obesity, history of snoring, type of surgery, airway management and type of payment) indicated by total effect of the DAG (S2 Fig) were included as the minimally sufficient adjustment set with the main exposure (PRE) in the final model (Table 6). Finally, the occurrence of PRE and mild to moderate but not severe PRE (vs non-PRE) were associated with higher excess hospital cost (OR = 1.4, 95% CI: 1.1, 1.6 / OR = 1.4, 95% CI: 1.1, 1.7, respectively) (Table 6).

**Table 5. Predictors of excess hospital cost, length of stay and prolonged length of stay (N = 312).**

| Variables | Geometric mean$^G$ of excess hospital cost, mean (GSD) | p-value | Length of stay, Mean (SD) | p-value | Prolonged length of stay, N (%*) | | p-value |
|---|---|---|---|---|---|---|---|
| | | | | | Yes (n = 35) | No (n = 277) | |
| PRE | | 0.003[a] | | 0.077 | | | <0.001[c] |
| Yes (n = 63) | 4315.64 (2.44) | | 0.70 (1.19) | | 15 (23.8) | 48 (76.2) | |
| No (n = 249) | 2951.30 (2.34) | | 0.41 (1.05) | | 20 (8.0) | 229 (92.0) | |
| Severity of PRE | | 0.005[b] | | 0.141 | | | <0.001[c] |
| Non-PRE (n = 249) | 2951.30 (2.34) | | 0.41 (1.05) | | 20 (8.0) | 229 (92.0) | |
| Mild to moderate (SpO$_2$ 86–99%) (n = 49) | 4536.90 (2.41) | | 0.74 (1.29) | | 10 (20.4) | 39 (79.6) | |
| Severe (SpO$_2$ < 86%) (n = 14) | 3604.72 (2.48) | | 0.57 (0.76) | | 5 (35.7) | 9 (64.3) | |
| Preoperative planned admission | | <0.001[a] | | <0.001[a] | | | <0.001[c] |
| Yes (n = 72) | 6836.29 (1.92) | | 1.89 (1.53) | | 27 (37.5) | 45 (62.5) | |
| No (n = 240) | 2540.21 (2.20) | | 0.04 (0.21) | | 8 (3.3) | 232 (96.7) | |
| Age (years) | | <0.001[a] | | 0.003[a] | | | 0.018[c] |
| ≤ 6 (n = 203) | 2697.28 (2.34) | | 0.33 (0.93) | | 16 (7.9) | 187 (92.1) | |
| > 6 (n = 109) | 4402.82 (2.25) | | 0.72 (1.30) | | 19 (17.4) | 90 (82.6) | |
| Obesity | | 0.904 | | 0.968 | | | 0.398 |
| Yes (n = 36) | 3165.29 (2.10) | | 0.47 (1.18) | | 2 (5.6) | 34 (94.4) | |
| No (n = 276) | 3197.10 (2.41) | | 0.46 (1.07) | | 33 (12.0) | 243 (88.0) | |
| Upper respiratory tract infection | | 0.632 | | 0.344 | | | 0.780 |
| Yes (n = 35) | 3394.8 (2.16) | | 0.63 (1.52) | | 3 (8.6) | 32 (91.4) | |
| No (n = 277) | 3165.29 (2.41) | | 0.44 (1.02) | | 32 (11.6) | 245 (88.4) | |
| History of snoring | | 0.767 | | 0.671 | | | 0.715 |
| Yes (n = 93) | 3261.69 (2.53) | | 0.51 (1.11) | | 9 (9.7) | 84 (90.3) | |
| No (n = 219) | 3165.29 (2.32) | | 0.45 (1.08) | | 26 (11.9) | 193 (88.1) | |
| Type of surgery | | <0.0001[b] | | 0.301 | | | 0.884 |
| Urology (n = 133) | 3677.54 (1.77) | | 0.38 (1.02) | | 13 (9.8) | 120 (90.2) | |
| ENT (n = 68) | 3533.34 (2.77) | | 0.40 (0.63) | | 9 (13.2) | 59 (86.8) | |
| Eye (n = 56) | 1978.31 (3.22) | | 0.70 (1.17) | | 7 (12.5) | 49 (87.5) | |
| Others (n = 55) | 3294.47 (2.12) | | 0.51 (1.50) | | 6 (10.9) | 49 (89.1) | |
| ASA classification | | 0.118 | | 0.038[b] | | | 0.004[c] |
| 1 (n = 102) | 2951.30 (2.27) | | 0.43 (1.09) | | 8 (7.8) | 94 (92.2) | |
| 2 (n = 193) | 3229.23 (2.41) | | 0.43 (0.94) | | 21 (10.9) | 172 (89.1) | |
| 3 (n = 17) | 4722.06 (2.56) | | 1.12 (2.09) | | 6 (35.3) | 11 (64.7) | |
| Type of GA | | 0.0004[b] | | 0.687 | | | 0.804 |
| GA alone (n = 201) | 2779.43 (2.72) | | 0.49 (1.07) | | 23 (11.4) | 178 (88.6) | |
| GA with caudal block (n = 74) | 4315.64 (1.58) | | 0.46 (1.17) | | 9 (12.2) | 65 (87.8) | |
| GA with PNB (n = 37) | 3827.63 (1.55) | | 0.32 (0.97) | | 3 (8.1) | 34 (91.9) | |
| Induction agents | | 0.015[a] | | 0.008[a] | | | <0.001[c] |
| Sevoflurane (n = 266) | 3010.92 (2.29) | | 0.34 (0.77) | | 22 (8.3) | 244 (91.7) | |
| Propofol (n = 46) | 4447.07 (2.69) | | 1.17 (2.01) | | 13 (28.3) | 33 (71.7) | |
| Airway management | | <0.001[b] | | 0.001[b] | | | 0.011[c] |
| Mask (n = 52) | 972.63 (1.82) | | 0.02 (1.40) | | 0 (0) | 52 (100) | |
| LMA (n = 146) | 3944.19 (1.73) | | 0.45 (1.05) | | 17 (11.6) | 129 (88.4) | |
| ETT (n = 114) | 4188.09 (2.36) | | 0.69 (1.29) | | 18 (15.8) | 96 (84.2) | |
| Narcotic use | | <0.001[b] | | 0.281 | | | 0.488 |
| None (n = 5) | 566.80 (1.55) | | 0 (0) | | 0 (0) | 5 (100) | |

*(Continued)*

**Table 5.** (Continued)

| Variables | Geometric mean[G] of excess hospital cost, mean (GSD) | p-value | Length of stay, Mean (SD) | p-value | Prolonged length of stay, N (%*) | | p-value |
|---|---|---|---|---|---|---|---|
| | | | | | Yes (n = 35) | No (n = 277) | |
| IV fentanyl (n = 297) | 3229.23 (2.34) | | 0.46 (1.07) | | 33 (11.1) | 264 (88.9) | |
| Caudal narcotic (n = 10) | 5597.08 (1.79) | | 0.90 (1.60) | | 2 (20.0) | 8 (80.0) | |
| Type of payment | | 0.012[b] | | 0.025[b] | | | 0.124 |
| UC (n = 214) | 3468.38 (2.41) | | 0.49 (1.10) | | 26 (12.1) | 188 (87.9) | |
| CGD (n = 54) | 2864.07 (2.23) | | 0.44 (1.06) | | 6 (11.1) | 48 (88.9) | |
| Self-pay (n = 37) | 2164.62 (2.23) | | 0.16 (0.55) | | 1 (2.7) | 36 (97.3) | |
| Government corporation (n = 5) | 5486.25 (2.72) | | 1.80 (2.49) | | 2 (40.0) | 3 (60.0) | |
| Private insurance (n = 2) | 5115.34 (1.21) | | 1.00 (0) | | 0 (0) | 2 (100) | |

* row percent

[a] $p < 0.05$ by unpaired t- test

[b] $p < 0.05$ by F test statistic

[c] $p < 0.05$ by Chi-square test

[G] exponential of the log of the excess cost, Others = mass excision/gastroscopy, ASA, American Society of Anesthesiologist; ENT, Ear-nose-throat; GA, General anesthesia; GSD, Geometric standard deviation; PNB, Peripheral nerve block; LMA, Laryngeal mask airway; ETT, Endotracheal tube; IV intravenous; UC, Universal coverage; CGD, Comptroller General's Department; PRE, Perioperative respiratory event; $SpO_2$, Oxygen saturation.

## Discussion

This study examined hospitalization and excess hospital cost between PRE children and non-PRE children in pediatric ambulatory surgery. Our results regarding the impact of PRE on hospitalization and excess hospital cost in outpatients surgery were consistent with our previous study focusing on inpatients surgery except for type of admission [7]. Since the ambulatory surgery in our setting is still developed, 25% of cases are preoperative planned admissions. Since 1 day was the average length of stay for preoperative planned admissions, our criteria for prolonged admission was different among planned ($\geq$ 2 days) and unplanned admission ($\geq$ 1 day) which was entirely different from our previous study in which the outcomes were any hospital stay and length of stay [7]. In the present study, we focused only on outpatients surgery in which prolonged hospital stay (yes / no) was more appropriate and provided a simpler interpretation when we encountered with different types of admission (planned / unplanned admission). Paine et al. [17] reported that the average length of hospital stay for good candidates for ambulatory cleft lip repair was 1 day, which was quite similar to the length of stay for planned admission patients in our setting. Overall, PRE was associated with prolonged hospital stay and higher excess hospital cost, i.e. accommodation, meals, laboratory expenses, anesthesia charge, and nursing care service (Table 3). Although medications, oxygen therapy and material charge were higher in the PRE group, they were not significantly different compared to our previous study [7] because this present study was confined to only ambulatory surgery cases who required less medication supplies.

Studies on oxygen desaturation ($SpO_2 < 95\%$) in the PACU have reported prolonged length of stay in the PACU [6, 18]. We found that the duration of anesthesia was longer in the PRE group compared to non-PRE group possibly resulting from the majority of PRE occurring in the intraoperative period (64%). Even though the average duration of a PRE event was quite brief (median 1–5 min), some patients required more time (> 30 min) to manage the PRE. Therefore, we considered duration of anesthesia as a consequence of PRE.

**Table 6. Multiple logistic regression by total effect model predicting relative probability of prolonged length of stay and log excess hospital cost (N = 312).**

| Main exposure variables | Prolonged length of stay[†] | |
| --- | --- | --- |
| | Adjusted odds ratio (95% CI) | p-value[*] |
| Planned admission and non-PRE | 1[a] | <0.001 |
| Planned admission and PRE | 1.69 (0.49, 5.78)[a] | |
| Unplanned admission and non-PRE | 0.011 (0.002, 0.07)[b] | |
| Unplanned admission and PRE | 0.26 (0.08, 0.87)[c] | |
| Planned admission and non-PRE | 1[a] | <0.001 |
| Planned admission and mild to moderate PRE | 1.22 (0.32, 4.63)[a] | |
| Planned admission and severe PRE | 7.56 (0.52, 110.0) [a] | |
| Unplanned admission and non-PRE | 0.011 (0.002, 0.067)[b] | |
| Unplanned admission and mild to moderate PRE | 0.19 (0.04, 0.81)[c] | |
| Unplanned admission and severe PRE | 0.51 (0.09, 2.81)[ac] | |
| **Main exposure variables** | **Log excess hospital cost[‡]** | |
| | Adjusted cost ratio (95% CI) | p-value[**] |
| PRE (ref = non-PRE) | 1.35 (1.13, 1.62) | 0.001 |
| Severity of PRE (ref = Non-PRE) | 1[a] | 0.004 |
| Mild to moderate | 1.39 (1.14, 1.68)[b] | |
| Severe | 1.23 (0.87, 1.73)[ab] | |

[*]Likelihood ratio test

[**]F-test. PRE, Perioperative respiratory event. ref: reference group.

[†]Minimally sufficient adjusted set: Age, American Society of Anesthesiologists classification, snore, type of surgery, type of payment.

[‡]Minimally sufficient adjusted set: Age, American Society of Anesthesiologists classification, obesity, snore, type of surgery, airway management, type of payment. Odd ratios within columns and variables that have no superscript ([abc]) in common differ significantly at p <0.05 (Wald test).

## DAG to reduce potential bias in the relationship between PRE and prolonged hospital stay post-surgery

Because some risk factors may be associated with both PRE and hospital stay post-surgery, we used the total effect of the DAG method to identify biasing pathways that needed to be blocked by including the potential confounders into the final model for prolonged length of stay [16, 19]. The minimally sufficient adjustment set indicated by the DAG (S1 Fig) among the main exposure and outcome consisted of age [11], history of snoring [3], ASA classification [20], type of surgery [21, 22] and type of payment [23, 24]. When using a cross-classification variable (preoperative planned/unplanned admission and PRE/non-PRE), the occurrence of PRE in a preoperative unplanned admission was associated with 24-fold increased odds of prolonged hospital stay. Since other potential confounding variables (in the minimally sufficient adjustment set) were not the main exposure, the result of association between those confounders and outcome by total effect of the DAG were omitted. Among non-PRE cases, preoperative unplanned admission decreased the risk of prolonged hospital stay (0.01-fold) compared to preoperative planned admission. If unplanned admission patients had no perioperative complications, they can be discharged home after surgery.

Although most studies have reported the incidence and predictors of prolonged stay and unplanned admission in ambulatory surgery in children [25, 26] and adults [27], we discovered a significant effect modification between preoperative unplanned admission and having PRE in the model of prolonged hospital stay in pediatric ambulatory surgery. When we looked

at the severity of PRE in a preoperative unplanned admission, the more severe PRE ($SpO_2$ <86%) was associated with a higher odds of prolonged hospital stay (OR 46.4, p <0.001). A meta-analysis of laparoscopic cholecystectomy in adults reported that the unplanned admission rate in ambulatory surgery was comparable with the prolonged hospitalization of inpatients [28].

## DAG to reduce potential confounders in the relationship between PRE and excess hospital cost

According to S2 Fig, potential biasing factors were age, obesity, history of snoring ASA classification, type of surgery, airway management and type of payment. Some studies reported that use of a face mask or laryngeal mask airway compared with tracheal intubation significantly decreased the risk of respiratory complication in pediatric anesthesia [29–31] which might lessen the cost of hospitalization. Since other potential confounding variables (in the minimally sufficient adjustment set) were not the main exposure, the result of association between those confounders and outcome by total effect of the DAG were omitted. PRE increased the odds of excess hospital cost 1.35-times when compared to non-PRE. When we looked at the severity of PRE, mild to moderate PRE increased the odds of excess hospital cost by almost 1.4 times. We conclude that PRE was associated with a 35–39% higher excess hospital cost regardless of planned or unplanned admission.

## Contribution margin and excess hospital cost

We used contribution margin (revenue minus variable costs) to describe the financial resources produced by hospital activities and found that the positive contribution margin is economically beneficial to pay for a hospital's fixed costs [14]. Therefore, the contribution margin was compared between PRE and non-PRE. The mean differences in contribution margin (per patient) (PRE minus non-PRE) was 1,523 baht in overall operation, whereas they were higher (2,279 baht) in direct laryngoscopy and bronchoscopy operation. This result implies that having ≥1 PRE in ambulatory surgery was associated with a higher hospital charge and more hospital direct cost compared to non-PRE. However, this contribution margin result came from univariate analysis comparing between groups of having at least 1 PRE and groups not having PRE which was confirmed by multivariate analysis of excess hospital cost (Table 6).

The association of PRE with higher excess hospital cost was likely due to higher variable cost (accommodation, meal, X-ray and laboratory) related to admission and higher anesthesia cost related to non-admission (Table 3). The geometric mean [geometric SD] of excess hospital cost was higher for patients whose costs were covered by the Universal Coverage Scheme (mean = 3,468 [2.41]) compared to self-pay (mean = 2,165 [2.23]) and the Comptroller General's Department (mean = 2,864 [2.23]) (p = 0.012, Table 5). We could not conclude that the type of payment was associated with excess hospital cost since we did not focus the multivariate analysis on type of payment and excess hospital cost. However, a majority of hospital charge was paid by the Universal Coverage Scheme (68.6%); if PRE occurs, the hospital will be responsible for the higher excess cost related to PRE.

## Hospital planning

According to hospital policy, we plan to expand our surgical day care service for both GA (by anesthesiologist) and local anesthesia (by surgeon) to have same day discharge for > 90% of the patients. Therefore, in the near future, any hospital stay will be specific to only unplanned admissions. Even if the event is only mild, PRE occurs quite often and PRE with unplanned admission was associated with 24-fold increased odds of prolonged hospital stay. PRE itself

regardless of unplanned admission can produce 35% higher excess hospital cost or an increase in differences in contribution margin of 1,523 baht (48.74 U.S. dollars) per patient in pediatric ambulatory surgery. Since most of our hospital costs are paid by the Universal Coverage Scheme based on diagnosis-related group weighting per case for non-PRE children, a hospital could lose 13% to 62% of the reimbursement if PRE occurs in pediatric ambulatory surgery. Thus, anesthesiologists have the important role of optimizing high-risk patients at the surgical day care clinic or selecting a good candidate for outpatients' surgery. Cancelling non-optimized cases (controversial respiratory symptoms) in advance before patients arrive at the hospital will reduce the risk of operating room cancellation and PRE occurrence (direct hospital cost) as well as the indirect cost of patient transportation [32, 33]. In cases where cancellation is not possible, the anesthesiologist can reduce the risk of prolonged hospital stay by early detection and prompt management of PRE to reduce hospital financial losses related to PRE.

## Strengths and limitations

There are several strengths of our study. First, this secondary data analysis focused on ambulatory surgery, demonstrating excess hospital costs and contribution margins that compared a group of patients with at least 1 PRE and a group of patients without PRE which has rarely been done before. Second, we used a DAG and multivariate model to appropriately reduce confounding. Even though we attempted to examine net revenue and contribution margin between PRE and non-PRE, the total margin, which represents hospital profit margin, was not examined [34]. However, this knowledge will activate public health sectors, especially hospitals in the Ministry of Health, to be aware of the risk of PRE in pediatric ambulatory surgery, which can impact hospital finances.

## Conclusions

PRE with unplanned admission in pediatric ambulatory surgery was associated with a 24-times increased odds of prolonged hospitalization post-surgery. PRE occurrence can result in a 35% higher excess hospital cost in non-cardiac surgery.

## Supporting information

**S1 Fig. Hypothesized causal relationship between perioperative respiratory events and hospital stay after adjusting for age, American Society of Anesthesiologists classification, snore, type of surgery, type of payment using directed acyclic graph.** PRE, Perioperative respiratory event; LOS, Length of hospital stay; Sx, type of surgery; Airway, Airway management; ASA, American Society of Anesthesiologists; Type of GA, Type of general anesthesia; Induct, Induction agent; URI, Upper respiratory tract infection.
(TIF)

**S2 Fig. Hypothesized causal relationship between perioperative respiratory events and excess hospital cost after adjusting for age, American Society of Anesthesiologists classification, obesity, snore, type of surgery, airway management, type of payment using directed acyclic graph.** PRE, Perioperative respiratory event; LOS, Length of hospital stay; Cost, Excess hospital cost; Sx, type of surgery; Airway, Airway management; ASA, American Society of Anesthesiologists; Type of GA, Type of general anesthesia; Induct, Induction agent; URI, Upper respiratory tract infection.
(TIF)

**S1 Data. OPD312.**
(CSV)

## Acknowledgments

The authors would like to thank Mr. Glenn Shingledecker and Assistant Professor Edward McNeil for proofreading the final manuscript.

## Author Contributions

**Conceptualization:** Maliwan Oofuvong, Alan Frederick Geater, Virasakdi Chongsuvivatwong, Thavat Chanchayanon.

**Data curation:** Bussarin Sriyanaluk, Boonthida Suwanrat, Kanjana Nuanjun.

**Formal analysis:** Maliwan Oofuvong, Alan Frederick Geater.

**Investigation:** Maliwan Oofuvong.

**Methodology:** Maliwan Oofuvong.

**Supervision:** Alan Frederick Geater, Virasakdi Chongsuvivatwong.

**Writing – original draft:** Maliwan Oofuvong.

**Writing – review & editing:** Alan Frederick Geater, Virasakdi Chongsuvivatwong, Thavat Chanchayanon, Bussarin Sriyanaluk, Boonthida Suwanrat, Kanjana Nuanjun.

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
