## [Decision Letter · Decision Letter 0]

8 Apr 2021

PONE-D-21-08611

Does Perioperative Respiratory Event Increase Length of Hospital Stay and Hospital Cost in Pediatric Ambulatory Surgery?

PLOS ONE

Dear Dr. Oofuvong,

Thank you for submitting your manuscript to PLOS ONE. After careful consideration, we feel that it has merit but does not fully meet PLOS ONE’s publication criteria as it currently stands. Therefore, we invite you to submit a revised version of the manuscript that addresses the points raised during the review process.

We look forward to receiving your revised manuscript.

Kind regards,

Tai-Heng Chen, M.D.

Academic Editor

PLOS ONE

Reviewers' comments:

Reviewer's Responses to Questions

**Comments to the Author**

1. Is the manuscript technically sound, and do the data support the conclusions?

Reviewer #1: Yes

Reviewer #2: Yes

Reviewer #3: Yes

2. Has the statistical analysis been performed appropriately and rigorously? 

Reviewer #1: Yes

Reviewer #2: Yes

Reviewer #3: Yes

3. Have the authors made all data underlying the findings in their manuscript fully available?

Reviewer #1: Yes

Reviewer #2: Yes

Reviewer #3: Yes

4. Is the manuscript presented in an intelligible fashion and written in standard English?

Reviewer #1: Yes

Reviewer #2: Yes

Reviewer #3: Yes

5. Review Comments to the Author

Reviewer #1: The authors present a subgroup analysis of a single-institution, 13-month prospective cohort study of the effects of a perioperative respiratory event (PRE) on length of stay and hospital cost, in children having ambulatory surgery. Of note, the authors' ref. [7] is their prior study that focused mostly on inpatients, and they state that the purpose of this manuscript's subgroup analysis is to investigate these effects on ambulatory pediatric surgery patients.

The authors employ directed acyclic graphs to determine which variables to include in their multivariable regression models. They then find that PREs with unplanned admission were associated with an increased length of stay and increased hospital cost.

The paper is a reasonable extension of the authors' prior work. There are some issues that should be addressed before publication:

1. The authors' explanation of the calculation of the contribution margin (lines 126-140) is not clear. It seems that they use the equation

(contribution margin) = 0.6 x (hospital charges)

based on assumptions from their prior work. This discussion should be clarified, as this calculation seems to be critical to one of the main conclusions - that increased costs to patients were associated with PRE.

2. Why are there so few general surgery procedures reported? The procedures in Table 4 include ophthalmology, otolaryngology, urology and gastroenterology procedures. While excision of superficial mass and hydrocelectomy (which often includes inguinal hernia repair) are included, the authors should comment on this. I would expect there to be ambulatory hernia repairs, cholecystectomies, gastrostomy placements, and the like.

3. The results from this study should be compared in more detail to those of the authors' ref. [7] in the Discussion.

4. The manuscript needs to be proofread again.

Reviewer #2: It is an interesting paper that provides information on little-studied aspects of ambulatory surgery in children.

I am struck by the fact that 27.1% of the patients were eliminated for not being able to contact their parents/guardians. Seems too high for ambulatory procedures where children get to the hospital accompanied by someone.

It would be convenient for the authors to explain their thinking about the relationship between PRE and the ASA classification of patients.

Line 217 has some writing problems.

On line 332 the authors write about the excess hospital cost. Here, it seems to me that the variables medications and oxygen therapy should not be included in the paragraph because both variables did'n reach statistical significance.

Statistically it has some tables that are difficult for the regular reader to understand, but they are finally understood with the specifications of each table.

Reviewer #3: This is an important paper despite no surprising findings. I have almost none criticism; the number of partcipiants seems short to overcome the effect of so many covariables. Try to use more accessible and simpler language. What is the explanation for a lower odds ratio in case of unplanned admission and non-PRE? The inclusion of "planned admission" may question the external validity of the conclusions. Is it possible to apply the same methodology excluding "planned" admissions", a more ubiquitous reality in pediatric ambulatory surgical centers?

6. PLOS authors have the option to publish the peer review history of their article (what does this mean?). If published, this will include your full peer review and any attached files.

Reviewer #1: No

Reviewer #2: No

Reviewer #3: No

---

## [Author Response · Author response to Decision Letter 0]

15 Apr 2021

Dear PLOS ONE Editor,

We would like to resubmit a manuscript entitled “Does Perioperative Respiratory Event Increase Length of Hospital Stay and Hospital Cost in Pediatric Ambulatory Surgery?

” as an original article in your journal.

We are thankful to have a chance to revise our manuscript. To address the editor’s comment, we have deposited our protocol in protocols.io. We have also highlighted the changes in the current version. Attached is a point-by-point response to the reviewer’s concerns. 

Reviewers' comments:

Comments to the Author

Reviewer #1: The authors present a subgroup analysis of a single-institution, 13-month prospective cohort study of the effects of a perioperative respiratory event (PRE) on length of stay and hospital cost, in children having ambulatory surgery. Of note, the authors' ref. [7] is their prior study that focused mostly on inpatients, and they state that the purpose of this manuscript's subgroup analysis is to investigate these effects on ambulatory pediatric surgery patients.

The authors employ directed acyclic graphs to determine which variables to include in their multivariable regression models. They then find that PREs with unplanned admission were associated with an increased length of stay and increased hospital cost.

The paper is a reasonable extension of the authors' prior work. There are some issues that should be addressed before publication:

1. The authors' explanation of the calculation of the contribution margin (lines 126-140) is not clear. It seems that they use the equation

(contribution margin) = 0.6 x (hospital charges)

based on assumptions from their prior work. This discussion should be clarified, as this calculation seems to be critical to one of the main conclusions - that increased costs to patients were associated with PRE.

Response: Thank you for your comments. It is correct that contribution margin was simply calculated from 0.6 x hospital charges, in our setting since we used a cost-to-charge ratio of 0.4 based on our previous study as well as omitting the fixed cost. We added this information in the Financial information subsection of the Methods section (line 139-140, page 7). 

2. Why are there so few general surgery procedures reported? The procedures in Table 4 include ophthalmology, otolaryngology, urology and gastroenterology procedures. While excision of superficial mass and hydrocelectomy (which often includes inguinal hernia repair) are included, the authors should comment on this. I would expect there to be ambulatory hernia repairs, cholecystectomies, gastrostomy placements, and the like.

Response: Hernia repair (herniotomy) was grouped in the minor urologic surgery category. We added this information in the legend of Table 4. We do not have ambulatory cholecystectomies in children, usually they are adult cases and they are usually admitted (inpatients) before surgery because of fever and abdominal pain. Gastrostomy placements patients in our setting are mostly cerebral palsy patients that are usually inpatients. 

3. The results from this study should be compared in more detail to those of the authors' ref. [7] in the Discussion.

Response: We have added this information as suggested in the discussion (line 328-332, page 24, line 334-338, 342-345, page 25).

4. The manuscript needs to be proofread again.

Response: The manuscript has been proofread by an English native speaker (Asst. Prof. Edward McNeil).

Reviewer #2: It is an interesting paper that provides information on little-studied aspects of ambulatory surgery in children.

Response: Thank you

I am struck by the fact that 27.1% of the patients were eliminated for not being able to contact their parents/guardians. Seems too high for ambulatory procedures where children get to the hospital accompanied by someone.

Response: Since it was a prospective cohort study, we could not obtain written informed consent from all outpatients at a given time, so the study was confined to consenting parents.

It would be convenient for the authors to explain their thinking about the relationship between PRE and the ASA classification of patients.

Response: Even though Table 2 shows significant difference of ASA classification between PRE and non-PRE group, the ASA classification 3 was similar between the PRE (4.8%) and non-PRE group (5.6%). A higher ASA classification could easily develop PRE; therefore, ASA classification was one of the adjusted variables in both prolonged admission and excess cost models. We add ASA detail in the Results section (line 209-212, page 11)

Line 217 has some writing problems.

Response: Thank you. We have rewritten it as “Approximately 40% of children in the PRE group were admitted for at least 1 day”. (line 217-218, page 11)

On line 332 the authors write about the excess hospital cost. Here, it seems to me that the variables medications and oxygen therapy should not be included in the paragraph because both variables did'n reach statistical significance.

Response: Thank you. We have rewritten the sentence in the discussion (line 342-345, page 25).

Statistically it has some tables that are difficult for the regular reader to understand, but they are finally understood with the specifications of each table.

Response: I apologize for that. I have added definitions of the specific terms in the manuscript (line 139-140, page 7, line 265-266, page 16) and in the legend of Table 4 (contribution margin) and Table 5 (geometric mean). 

Reviewer #3: This is an important paper despite no surprising findings. I have almost none criticism; the number of participants seems short to overcome the effect of so many covariables. 

Response: Thank you. We calculated the sample size to make sure the number of subjects in the study were adequate to reach our hypothesis.

Try to use more accessible and simpler language. 

Response: We apologize for that. We have added some definitions of specific terms in the manuscript which hopefully makes it easier to read. 

What is the explanation for a lower odds ratio in case of unplanned admission and non-PRE?

Response: Among non-PRE, preoperative unplanned admission decreased the risk of prolonged hospital stay compared to preoperative planned admission, which is basically true. If there were no perioperative complications occurring among preoperative unplanned admissions, they can be discharged home after surgery. We added this information in the discussion (line 365-368, page 26).

The inclusion of "planned admission" may question the external validity of the conclusions. Is it possible to apply the same methodology excluding "planned" admissions", a more ubiquitous reality in pediatric ambulatory surgical centers?

Response: Excluding preoperative planned admission out of the analysis process may be the ideal option. However, since we used secondary data with a limited sample size, we included an interaction term between type of admission and PRE, which is an alternative solution to reduce the type of admission bias.

Should you have more comments, please don’t hesitate to contact us

Thank you for your attention to our paper

Sincerely yours,

Assoc. Prof. Maliwan Oofuvong

Alan Frederick Geater

Prof. Virasakdi Chongsuvivatwong 

Assoc. Prof. Thavat Chanchayanon 

Bussarin Sriyanaluk 

Boonthida Suwanrat 

Kanjana Nuanjun

---

## [Decision Letter · Decision Letter 1]

27 Apr 2021

Does Perioperative Respiratory Event Increase Length of Hospital Stay and Hospital Cost in Pediatric Ambulatory Surgery?

PONE-D-21-08611R1

Dear Dr. Oofuvong,

We’re pleased to inform you that your manuscript has been judged scientifically suitable for publication and will be formally accepted for publication once it meets all outstanding technical requirements.

Kind regards,

Tai-Heng Chen, M.D.

Academic Editor

PLOS ONE

Reviewers' comments:

Reviewer's Responses to Questions

**Comments to the Author**

1. If the authors have adequately addressed your comments raised in a previous round of review and you feel that this manuscript is now acceptable for publication, you may indicate that here to bypass the “Comments to the Author” section, enter your conflict of interest statement in the “Confidential to Editor” section, and submit your "Accept" recommendation.

Reviewer #1: All comments have been addressed

Reviewer #2: All comments have been addressed

Reviewer #3: All comments have been addressed

2. Is the manuscript technically sound, and do the data support the conclusions?

Reviewer #1: Yes

Reviewer #2: (No Response)

Reviewer #3: Yes

3. Has the statistical analysis been performed appropriately and rigorously? 

Reviewer #1: Yes

Reviewer #2: (No Response)

Reviewer #3: Yes

4. Have the authors made all data underlying the findings in their manuscript fully available?

Reviewer #1: Yes

Reviewer #2: (No Response)

Reviewer #3: Yes

5. Is the manuscript presented in an intelligible fashion and written in standard English?

Reviewer #1: Yes

Reviewer #2: (No Response)

Reviewer #3: Yes

6. Review Comments to the Author

Reviewer #1: All my concerns have been met:

1. Authors clarified the calculation of contribution margin.

2. Authors clarified how surgical cases were classified as well as the pattern of general surgery cases at their institution.

2. Authors expanded discussion of the relation of the current work to their prior study.

4. Authors had the manuscript proofread by a native English speaker for clarity.

Reviewer #2: My previous comments have been answered with all satisfaction and it seems to me that, for my part, it can be published, although there are still flaws in the writing in English.

Reviewer #3: Despite the aa have failed to explain the rationale for a lower odds ratio in case of unplanned admission and non-PRE, I have no additional crticism. So "it is better to operate on unplanned admissions".

7. PLOS authors have the option to publish the peer review history of their article (what does this mean?). If published, this will include your full peer review and any attached files.

Reviewer #1: No

Reviewer #2: **Yes: **Eduardo Bracho-Blanchet

Reviewer #3: No

---

## [Editor Report · Acceptance letter]

3 May 2021

PONE-D-21-08611R1 

Does Perioperative Respiratory Event Increase Length of Hospital Stay and Hospital Cost in Pediatric Ambulatory Surgery? 

Dear Dr. Oofuvong:

I'm pleased to inform you that your manuscript has been deemed suitable for publication in PLOS ONE. Congratulations! Your manuscript is now with our production department. 

Kind regards, 

on behalf of

Dr. Tai-Heng Chen 

Academic Editor

PLOS ONE